# Living Interfaith Dialogue during the Lockdown: The Role of Women in the Italian Case

**Andrea Casavecchia *** , **Chiara Carbone *** and **Alba Francesca Canta ***

Department of Science of Education, University of Roma Tre, 00185 Roma, Italy

* Correspondence: andrea.casavecchia@uniroma3.it (A.C.); chiara.carbone@uniroma3.it (C.C.); albafrancesca.canta@uniroma3.it (A.F.C.)

**Abstract:** The aim of this article is to present some results of a study conducted in Italy exploring interfaith dialogue during and after the 2020 lockdown. To continue ritual practices during the COVID-19 emergency, several religious communities rethought all their forms of communication. They have shifted their activities mainly to the internet. Often, these transformations changed the forms of religious practices, but did not alter pre-existing cosmologies and theologies. How has physical distance affected interfaith dialogue, and what role did women play? To answer these questions, the research used semi-structured interviews with key informants, including opinion leaders of religious communities and experts. The analysis of the interviews paid special attention to the active role played by women in building bridges between different religions.

**Keywords:** dialogue; interreligion; woman; COVID-19; lockdown

## 1. Introduction

The article explores two related topics: the experiences of interreligious dialogue during and after the COVID-19 emergency, and the role played by women in promoting it. The pandemic containment measures adopted to limit infections in Italy and Europe (Ferguson et al. 2020) imposed a strict lockdown in 2020, which continued in 2021 in a less coercive way. The life of citizens, and civil and religious communities, have been severely conditioned and revolutionized for a long time; people have established new behaviours in everyday life inspired by prudence (Brown 2020). In some cases, the pandemic context has favoured the spread of fake news, such as the direct accusation against a Muslim minority of having spread the contagion in India (Cangemi 2021), or as happened in the forms of anti-Semitic hate speech on social networks in Italy (Pasta et al. 2021). Fortunately, the trend in Italy has been extremely contained and has not caused fractures between the various religions. A much-debated topic in the country was the relationship between religious freedom and restrictions on COVID-19. The topic was ridden by the No-Vax movement, but did not find support from the institutions of the various religions (Consorti 2020).

In fact, in this period, all the religions have supported the rules that imposed restrictions on citizens' mobility, respect for physical distancing, lockdowns, and a ban on people gathering in crowds. Across the world, the systems of measures taken by governments to fight infection called religious communities to modify their rituals, practices, liturgy and so on. All communications and meetings have been transferred to the web. This shift has changed the forms of practices, but did not transform the pre-existent worldviews, their cosmologies, or their theologies (Lorea et al. 2022). In some cases, the religious experiences have been creative and innovative (Neumaier and Klinkhammer 2022): the use of media there is not new, but the COVID-19 pandemic intensified their applications and types of presences on the web. At the same time, there has been a reconfiguration of sacred time and space (Neumaier and Klinkhammer 2022). In other cases, the more faithful were not ready for such changes, especially in traditionalist movements (Rauf 2022). The new situation has asked for reimagining belonging and participation in religious communities; the online

experiences were essential, but not an alternative for some liturgies that call for presence, such as the liturgy of the Eucharist for Catholics (Parish 2020).

In a time of crisis, experiences of interreligious dialogue have been promoted at a global level with different events: an example is the "Day of Prayer for Humanity", joined by Christians, Muslims, Hindus, Jews, Buddhists, and even atheists and agnostics (Corpuz 2021). Different religions together have played a specific role in interpreting human suffering and the meaning of life: this is one of the multiple dialogue experiences (Lehmann 2021) that create an open social space to build a culture of encounter. The experiences show four different interfaith and interreligious forms that should have helped humanity in the pandemic crisis: a dialogue of life, to allow people of different religions to live in a neighbourly spirit; of action, to build a human society; of theological exchange, to the understanding of religious world; and the religious experiences to share how to search for God (Corpuz 2021).

This article bases its analysis on semi-structured interviews with key informants followed up for a case study on interreligious dialogue in Italy (Canta et al. 2022), which was the first European country to experiment with containment measures. The interviewees are opinion leaders belonging to different religious communities, all of them involved in promoting dialogue between religions. They are engaged both institutionally and informally. At first, we deepen the concept of dialogue, its interreligious dimension and the role played by women. Then, we describe the research method adopted in the Italian context. Finally, we observe how the interreligious dialogue has continued and what contribution women have made in this field.

## 2. How Can We Define Dialogue?

The perspectives to observe dialogue are different. One concerns the dimensions of the public space and intertwines religious freedom and secularism; another dimension concerns the discussion between experts (i.e., theologians) comparing different doctrines; another looks at people engaged in building bridges between realities and communities of various religious affiliations; and yet another one relates to the dialogue between believers of different religions.

In any case, dialogue requires encounter and confrontation. It becomes principle in the relationship I-You, when the persons involved respect the unique identity of their dialogue partner (Buber 2014; Melnik 2020). Everyone discovers his/her own self and, at the same time, in the face-to-face relationship, overcomes the focus on identity to open up to otherness, which gives responsibility within the process of rèflet (Levinas 1990). Paul Ricoeur (1999) enhances in the dialogue the practice of self-recognition that drives the discovery of one's own identity through confrontation with the other. Moreover, dialogue becomes a space where the protagonists are involved in an egalitarian relationship and in a personal and communitarian transformation: from this involvement, they learn (Flecha 2000). Seyla Benhabib (2002) indicates that the process of interaction involves recognizing the other and their inviolable dignity, richness and diversity.

In dialogue, people can interpret their roles in different ways. They can confront each other in a depth path to learn more about the topic of the conversation; alternatively, they can speak in a spontaneous form of chat without a precise narrative thread with the simple purpose of spending time together (Gadamer 2004). In the first case, dialogue is a method for seeking the truth, while in the second case it is a tool for staying together.

Panikkar (1978) pays attention to some peculiarities of interreligious dialogue. Firstly, he highlights the specific religious nature of dialogue: this makes it impossible to suspend one's faith because it is precisely the religious worldview foundation. The different ways of believing open up the encounter between the inner consciousness of the dialogues (Panikkar 1978). Secondly, religious dialogue is: (1) polyglot, because it requires learning the language with which the interlocutors express themselves; (2) is political, because it occurs in public space, and it is practical when it links human experience to theory; (3) is

integral because it involves people in their entirety; and finally, (4) is continuous because it is constitutively imperfect since it is open to novelty (Panikkar 2000).

The dialogue paths have several features useful to detect some operational dimensions that constitute their essence. First of all, dialogue is an inner question (Panikkar 2000) that begins when certainties are questioned: dialogue fights fanaticism temptations and presumes "the engagement of people with critical minds, who question the obvious and also allow others to challenge them" (Moyaert 2013, p. 206). Dialogue calls for openness: in this sense, Hedge (2007) considers the game and the horizon as two features of interreligious dialogue. The first occurs in the interactions, when, entering into communication, the protagonists establish and create new—virtual—rules compared to those followed in everyday life. The second refers to the limit the gaze reaches and places new boundaries that expand when confronted with others. At last, dialogue is a method that enables the human agency: "interreligious dialogue possesses several promising benefits and potential that can benefit its participants and the social contexts where it takes place, such as providing mutual understanding, gaining knowledge of one's own and the others' religion and beliefs, reducing prejudice and conflict, and improving social coexistence" (Campdepadrós-Cullell et al. 2021, p. 189).

Women of faith offer a specific contribution to interreligious dialogue: to break down borders to join different communities (Ben-Lulu and Feldman 2022) or to build peace after war (Toki et al. 2015). Their actions are often underestimated for at least three reasons (Abu-Nimer 2015): (1) religious institutions are still structurally dominated by men; (2) interfaith dialogue is often defined as a space to talk about political problems or community relations problems in a society dominated by the patriarchal model; and (3) the few funders of dialogue actions neglect the role of women. They live a double marginalisation: on the one hand, they experience the closure of the patriarchal model dominant in religious institutions, and on the other, they experience the distrust of feminist movements (Nyhagen 2019). Even in Italy, they are considered complicit in the patriarchal system (Giorgi 2021). However, there is a growing presence of female believers in international feminist movements and creative groups that problematise the themes of religious institutions in relation to different arguments such as gender and sexuality (Knibbe and Bartelink 2019). Feminist currents have established themselves in different religions. They have posed themes of overcoming the masculine approach to the human in search of a dialogue between feminist theories and feminist theology (Eriksson 1995). They advance requests for a radical renewal of their communities on a theological and practical level to re-evaluate human identity from queer theory to overcome patriarchal culture (Bettie 2006) to combat an oppressive reading towards women, as observed by Amina Wadud (Rahemtulla 2017). Within a narrow space of action, women can open up innovative paths. They also experience interreligious dialogue as a way to change their religion (Zwissler 2012).

## 3. Methods

The technique of semi-structured interviews was used to achieve the case study objectives. The key informants were selected between opinion leaders from different and more representative religious communities in Italy and engaged in religious dialogue.

During pandemic distancing it was not possible to meet the key informants in person; thus, the interviews were conducted using video conferencing software (Teams, Zoom, Skype) and videotaped. This reflected the notion that, "researchers must be innovative in their use of digital technologies for the application of qualitative analysis" (Baker et al. 2020, p. 366). Although "remote" interviews do not offer the same quality as face-to-face ones, the specificity of the interviewees made it possible to collect information effectively (Thunberg and Arnell 2021). They were all recognized characters and used to speaking in public without impediments or difficulties in answering the questions. At the same time, for us researchers, it was more difficult to establish empathy with interviewees. In addition, sitting in front of a screen for over 1 h makes the interview extremely tiring.

However, since the site of the interviews was their own home, everyone chose their comfort zone and found themselves in a reassuring environment.

Although 30 individuals were invited to participate in the research (including two Buddhists and two Evangelical churches), only sixteen key informants responded (Table 1). Some interviewees belonged to religious communities, while others were scholars and people engaged in interreligious dialogue in the country. The decision to choose opinion leaders involved in interreligious dialogue during the lockdown derived from two reasons: they could be identified more easily, and their commitment made them reliable key informants in their communities and in other religious communities.

**Table 1.** Respondents, religious affiliation and their role.

| Code | Belonging/No Belonging to Religion/Self-Definition | Gender | Role of the Interviewee |
|---|---|---|---|
| 1 | Atheist | W | Co-editor of Micromega magazine (Magazine Co-editor) |
| 2 | Catholic | W | Founder and President of the Catholic Association "Women for the Church" (AWC—Founder) |
| 3 | Catholic | W | Religion for Peace Women's Coordinator for Italy (RP Coordinator) |
| 4 | Atheist | W | Union of Atheists and Agnostics—Former UNAA president (UAA—former president) |
| 5 | Catholic | W | Interfaith Center for Peace President (Cipax—President) |
| 6 | Muslim (Sunnis) | M | President of the Tiber Association (TA—President) |
| 7 | Researcher | W | Psychoanalytic Institute for Social Research (PIRS—member) |
| 8 | Tathata Vrindham | W | President Non-profit organization Tathata Vrindham International (TVI—President; ERPN—Coordinator) |
| 9 | Jewish | M | Union of Young Italian Jews President (UGEI—president) |
| 10 | Jewish | W | Reform Jewish Coordinator (RJ—Coordinator) |
| 11 | Catholic | M | Ecumenical Activities Secretariat (SAE—member) |
| 12 | University | M | Expert jurist on interreligious dialogue and secularism (Jurist) |
| 13 | Bahai | M | Coordinator of the Public Relations Office of the Bahá'í Community of Italy; (BCI—PR Coordinator) |
| 14 | Lutheran Christian | W | Member of the Ecumenical Council of the Churches of Geneva (ECCG—Member) |
| 15 | Muslim (Sunnis) | W | Member of Religion for Peace |
| 16 | Catholic | M | Office for interfaith and interreligious dialogue- Italian Episcopal Conference (IEC) |

The interviewees represent the more long-standing or ancient communities in Italy (Pace 2013a). The country's religious context, although characterized by a large majority of Catholics, is very varied: aside from Catholics, there are communities of Muslims, other Christian communities (such as Evangelicals and Orthodox), Buddhists, Hindus, and Sikhs.[1]

The semi-structured interview format revolved around the following topics: (1) the condition of religious dialogue in the country; (2) the difficulties experienced in the communities to which they belong; (3) the difficulties encountered in continuing the interreligious dialogue paths; (4) the relationship with the state and some key issues for one's own and other religions; and (5) the personal experiences of engagement. During the interview, particular attention was paid to the role of women regarding their distinguishing capacity in the construction of bridges between different religions (Canta 2019). To interpret the answers to the interviews, an epistemological approach was adopted to understand the social construction of the observed phenomena (Della Porta 2014). Through a con-

tent analysis framework, the statements collected by the interviewees were analyzed (Cipriani et al. 2021). The choice to use the semi-structured interview also served to enhance the relationship between the researcher and the interviewee. This technique allows us to approach the other to share the same topic(s) of interest, observed by the first and put into practice by the second.

## 4. Results

The results of our research, that emerged from the analysis of our interviews, are divided into three main topics. The first section collects the observations of the interviewees on how their communities have continued to undertake dialogue during the pandemic period, the second analyses how different communities experienced the interfaith dialogue before and after the lockdown, and the third delves into the female point of view underlying the notion that women drove the change, even during the pandemic. We cannot categorise women's roles, but we can observe some common elements and characteristics of how our key informants narrated their experiences of dialogue during COVID-19.

### 4.1. Seek Out New Forms for Ancient Practices

Particular attention should be paid to intra-community and inter-community dialogue and religious freedom, difficult topics that suffered from forced isolation in a time of intense uncertainty. During this period, the various religious groups faced limitations not so much in their religious freedom as in the way this freedom was expressed, and needed to evolve (Mazurkiewicz 2021). In some religious communities in Italy, such as the Catholic ones, and amongst different age groups during the pandemic, religious feelings increased compared to the pre-COVID period (Zygulski 2020). The importance of the Word of God and prayer, and the need to give meaning to one's life at a time when everything that previously made sense had vanished, increased; religious practices at home such as prayer, reading the Gospel or devotions also increased (Zygulski 2020).

Also in our research, interviews with opinion leaders showed that something has changed: religious life has reorganized into a domestic space in which the boundary between private and public has been blurred, crossed and confused (Piela and Krotofil 2020). In response to the need to give continuity to one's spirituality, the communities employed social networks, despite many people using the internet for religious information and spiritual guidance even before the pandemic (Helland 2000). In this period, these instruments represented a bridge for dialogue within many communities and have made it possible to feel close in the distance and present in the absence (Parish 2020), despite the preponderant desire to return to life sharing. While technological tools have not replaced some fundamental practices within some religious communities, such as the Eucharist for the Catholic community or the Yajña for the Hindu, a new dialogue has emerged. All communities have found a way to maintain a certain constancy in a year that would otherwise have resulted in the absence and crumbling of community practices. Encounters and internal sharing represent the raw materials for the very existence of communities that would otherwise increase individualized experiences (Parish 2020). Internal communitarian dialogue is also the first step that subsequently can lead to interreligious dialogue: it enables the sharing of experiences and values within a community to open up to the understanding of others. Despite the difficulties, this online experience was perceived as an opportunity to intensify participation within the communities, especially for those not used to meeting online: the web-based mode allowed the involvement not only of those who are geographically distant, but also of those who had become distanced from the practices for various reasons. In this sense, the communities managed continuity in the dialogue between people who already knew each other, but, at the same time, new relationships arose thanks to online instruments.

> There were also benefits [ . . . ] the tools we are using in this period will not be forgotten, because a total return to the pre-pandemic, I do not think that it will be, in the sense that useful tools have also been rediscovered! We certainly noticed a rapprochement of people who, for various reasons, had lost sight of a bit [ . . . ] an increase in requests for interaction. The psychological distress and forced isolation have led some people to request to reconnect. (w, TVI—President; ERPN—Coordinator, tathata Vridham)

During the pandemic, many people approached and opened themselves to faith and prayer. At a time when the virus was spreading faster and faster, religious practices gave people security and, especially, hope (Zygulski 2020). In line with our research and as stated by some respondents of a study conducted in Poland (Kowalczyk et al. 2020), faith and spirituality play a key role in coping with the pandemic crisis and increasing the sense of protection and control. Other research (Puchalski et al. 2009) has shown how religious beliefs and practices are associated with health-related aspects, such as the ability to cope with crises and harsh times, how fear motivates religious believers, and, by the same token, religion mitigates fear. However, despite this revival in religious feeling during the pandemic, the believers felt more distant from religious institutions (Puchalski et al. 2009), as confirmed also in our research. As a member of the Catholic Church and President of the Women for the Church Association told us during the interview, they continued to use tools already adopted even during the pandemic (via crucis, masses via streaming, novenas, confessional apps—A.Q. Scott 2016)—but forgotten was the idea of working on the sense of community trying to keep the contacts alive, to feel close in such a dramatic and lonely time. She said:

> It seems to me as if we had had the opportunity of the century and had not exploited it. (It has been forgotten) that for some people the instrument of domestic liturgies was a beautiful discovery.

Initially, the Family Prayer Resource Book was launched by the Catholic Church during the first lockdown and represented a way to care for and support a home spirituality: acts such as homemade bread, broken together with the whole family, Gospel reading, candle lighting during Advent and/or Christmas, and blessing rites made it possible to become celebrants at home. It was an experience of empowerment and clericalisation of ecclesial institutions which was immediately forgotten after the lockdown.

By contrast, as our research shows, other communities have been able to take advantage of the pandemic situation and use technological tools never used before to keep their dialogue alive. The Union of Young Italian Jews, in an effort to involve the older and younger people of its community (18–35 y.o.), has renewed some practices. Their president told us:

> We have reinvented ourselves online very well, even with professional tools [ . . . ] also because the online attention is much, much lower [ . . . ]. We have tried to set up the activities in a certain way, since we have also addressed an audience of young people (m, UGEI—president, Jewish).

Short conferences, daily meditation sessions of 2–3 min, podcasts, innovative blogs, magazines: all simple ingredients for a better interreligious dialogue. It was a dialogue that, at the same time, opened to other communities thanks to Ask a Jew, a non-institutionalized blog aimed at everyone who wanted to know more about Judaism and to put all kinds of questions to the experts. A simple and innovative blog but, especially, an instrument aimed at all those eager for confrontation.

In addition to the various online dialogue experiences, more 'original' experiences have been promoted by other communities, such as the Muslim community in Rome, which has received new stimuli and adopted new modalities as a result of the COVID-19 restrictions. These restrictions turned everything upside-down, changed places, and physically stopped people. Mosques, as well as all other places of worship, were closed; none of the religious practices could continue. The restrictions had a direct impact on



the routines of all communities (World Health Organization 2021). "This is the first time that the Friday congregational prayers have been suspended across the region and the decision was taken after the region reported its first death on Thursday" (Alam 2021, n.p.). Many Muslim communities, in every country including Italy, were forced to suspend both Friday Prayer and Ramadan. How to deal with these moments? How to deal with such a sudden change? These are the questions we asked our opinion leaders from the Muslim communities, to which we can respond: as it has never been done before, online.

Friday Prayer used digital means to continue to give hope and open itself up to the presence of women and men, unlike in 'normal' times, when prayer celebrations become spaces dedicated "exclusively" to men since their presence, unlike that of women, is mandatory.

Another innovation concerned the modality of Eid al-Fitr, the breaking of the fast on the occasion of Ramadan, which needed to be organised in a different way. In the last ten days since restrictions were imposed (last ten days before Ramadan), the number of visitors to the mosques for prayers increased, and believers met more often than usual. In this sense, physicality and closeness play an essential role. Unable to celebrate a moment of group sharing via a physical group presence, a Muslim community used technology to create a virtual space that would, unlike the traditional ritual, include the participation of believers of other religious groups, as one interviewee told us (m, TA—President, Muslim). This moment turned into an act of dialogue (w, TVI—President; ERPN—Coordinator Tathata Vridham); intra- and inter-communitarian for many, and an opportunity to discuss various religious practices.

If the pandemic has challenged many of the certainties of human beings and brought many problems to the surface, it has certainly not alleviated the need for true relationships and dialogue within communities. On the contrary, they have often become more acute, and found different and creative forms of expression. All the communities have found a way to discuss such matters, both those who have already cultivated dialogue through digital tools, and those which reinvented themselves through them. The transition from physical to online has changed rituals, feelings, approaches and spirituality (Ben-Lulu 2021), but has not changed the desire to encounter, especially in terms of physical presences that remain the preferred way to conduct dialogue. For many communities, finding a new way to continue an online dialogue within them was also the key to opening up and cultivating a more intense interreligious dialogue. One of our interviewees gave us an example: not having support from her religious denomination (Catholicism), she turned her gaze towards activities organised by other religious communities, finding a new sense of spirituality in that pandemic period and feeling important (w, Catholic, AWC—Founder).

*4.2. Interfaith Dialogue in Italy*

In the same way, the interfaith dialogue that, during the Second Vatican Council (1962–1965) with the Dignitatis Humanae e Nostra Aetate Declarations, assumed a fundamental role in the Italian debate, suffered the same fate and had to change. Despite the previous online experience serving to maintain a certain continuity in the dialogue, many essential difficulties had to be faced. In particular, two problems arose from the analysis of our interviews: the problem of interreligious dialogue in relation to religious freedom in Italy during this crisis, and the issue of how interreligious dialogue continued during the pandemic.

First of all, of particular relevance is the issue of religious freedom in Italy that necessarily emerged during this period of emergency. Religious freedom is recognised as a constitutional right but lacks a framework of law and a passage regarding the different starting points of religions. Each community had different spaces of freedom, due to restrictions that created inequalities and difficulties in proceeding with the interreligious dialogues between each other, particularly for those communities who did not sign an agreement with the state (w, TVI—President; ERPN—Coordinator, Tathata Vridham).

> In Italy, there is religious freedom but there is no law on religious freedom. You exist because you have made an agreement with the Catholics, who are the absolute majority. And this is a very big problem because it is true that it gives advantages to those who have made understandings—all kinds of advantages, we know—but it becomes a limiting constraint for others who are not part of that group. (w, ECCG—Member, Lutheran Christian)

This situation has produced a difference in positions between communities in Italy, which remained latent before the pandemic, and a sense of revenge against the majority religion, harshly reported in particular by the atheist interviewees (w, UAA—former president), and also recognized by Catholics as an impediment to peaceful dialogue (m, IEC, Catholic). Despite the presence of Christian forms other than Catholicism or other religions, Italian culture continues to be imbued with Catholic values in several areas (Cesnur 2021). After the first lockdown, when the different places of worship began to open again, a different procedure emerged between communities that had an agreement with the state and those that did not. Many groups, therefore, felt excluded and received different treatment: for example, as our interviewee told us, the Hindu community that is part of Religion for Peace suffered from such treatment because, not having an agreement with the state, they did not benefit from the reopening of places of worship like other religious communities. This issue produced an advantageous position for the Catholic community compared to other religions. Such inequality does not favour dialogue and does not recognize the dignity of other faiths in many places, such as schools or in places of worship.

The issue of Catholic religious teaching (w, Cipax—President, Catholic), or the difficulty of allowing some communities to be able to pray in a holy place (i.e., the Muslim communities in Rome—m, IEC, Catholic), are only some examples of all the problems that need to be solved in Italy to guarantee the same freedom of religious expression to all religious groups. Thinking about the interfaith dialogue as a positive interaction between different religious traditions that cooperate and act for tolerance and mutual respect (Panikkar 1978), and considering the same space of action (in this case, Italy), if there are differences in the concrete freedom of expression between the groups, it results in a lack of positive interreligious dialogue. Dialogue can only be developed by starting from the right to be considered equally part of the social context.

> However, one cannot deprive a person of being able to express their faith either. A lot of work needs to be done on this in Italy. There are steps that must be taken. A very complex subject, but it must be addressed. (m, IEC, Catholic)

Concerning the second result that emerged on how interfaith dialogue has continued, it was affected by these differences in religious freedom and by the need to transform and reinvent physical practices in an online version during the crisis, even if many practices could not have been replaced.

> We started a project called Bridge which was really a project of interreligious dialogue [ . . . ] So many moments, for example, could not be replicated with digital tools: folk festivals that united different communities, traditional dance and singing performances that became a moment of sharing for all, trips that transformed dialogue into a true act of interreligiousness. (w, PISR—member, Catholic)

This instrument of interreligious dialogue necessarily suffered a severe limitation, as it could only organise online meetings, albeit in an innovative way (i.e., psychoanalytical reading of the Quran—PIRS member), and suffered a slowdown process that was only restarted two years later. There was the conviction that it is crucial for interfaith partners to work together at the frontlines and that their work be fully embedded in larger national and secular organisational strategies to provide relief for communities who are suffering (Thompson 2020). From the experiences of the opinion leaders interviewed, it emerged that the ways in which religious discourses and practices have been rearticulated vary from

community to community, just as new forms of dialogue and relations with the outside world have emerged, as this witness points out:

> Religion for Peace found a right way to relate. Besides the meeting, there were prayers in common: in common in the sense that each participated with their own religious culture in meetings on the subject, for example, of populations defrauded of their rights. Here, then, everyone reads his prayer, his psalm [ . . . ] and this too is interesting! (w, RP—Coordinator, Catholic)

However, the interviews revealed profound differences in the perception of the impact the pandemic has had on interreligious dialogue and the modalities of such dialogue (Ben-Lulu 2021). When the dialogue is not used to being cultivated online or when there are some moments that need to be conducted and lived in person, it can cool down:

> The religious community is undoubtedly based on physicality; on this, undoubtedly, the pandemic has cooled, especially inter-religious relationships and less relationships in individual communities. (w, UAA—former president, atheist)

On the other hand, the pandemic has brought people back to their religious practice—not physically, but according to spiritual dynamics, which recognise belonging to a system of values, symbols and beliefs. According to some interviewees, these tools have also been a stimulus to activate interreligious dialogue.

> Concerning interreligious dialogue today, the focus is on public action, that is, on the ability to create coalitions, alliances between different spiritual traditions, in order to face the challenges of today's society. There is also a lot of work with respect to mutual knowledge and the quest to make the divisive and aggressive phenomena resulting from non-knowledge less dangerous! [ . . . ] Collaboration means having a greater impact on the major issues in society. [ . . . ] Interfaith dialogue is a resource for inclusion and social cohesion. (w, TVI—President; ERPN—coordinator)

In this sense, the body becomes the vehicle and means by which social relations are constructed: what facilitates the constriction of the social bond is spatial proximity, since proximity (including in a physical sense) creates familiarity between people and connects mutual intentions and expectations. Through the body we are involved in a process of sensations, perceptions and relationships that give meaning, shape experience and help articulate social reality (Scheper-Hughes and Lock 1987). The concrete physicality of the human being and the body as medium also becomes a way to practise and build interreligious dialogue through travel—a possibility that the pandemic completely stopped.

> For my work, I asked myself: what can I do to improve the situation for dialogue and peace? At that time, I decided to make a trip every year to a Muslim country: Iran, Egypt, Morocco. And I discovered many positive initiatives for dialogue on the part of Muslims, I found many friends and hospitality [ . . . ] but you can only do this live, you cannot do it online, via digital or telephone. (w, ECCG—Member, Lutheran Christian)

One method that was employed to cultivate dialogue was to take trips around the world to know and deeply understand other cultures, their religious beliefs, and their symbols and practices. On the one hand, technological tools can help us keep in touch in a moment of distance; on the other, they cannot supplant these acts of dialogue. There are experiences of dialogue that cannot be reproduced and translated in the same way simply by changing the scenario in which they take place. The field of presence differs from the digital one: both can integrate but not replace, coexist but not isolate. Two spaces of placement have different foundations, which presuppose specific practices and modes of dialogue, a new space of redefinition. In this space, dialogue becomes the principal means for pluralism and inclusiveness and requires a guarantee of freedom of expression and the creation of a ground for coexistence. Moreover, this space "is somehow like a manoeuvring

space, which should not be a neutral space. It should not be an aseptic space, it should be an inclusive space" (w, PIRS—member, Catholic).

If, on one hand, the physicality of the dialogue which fosters the relations has temporarily vanished during the pandemic in favour of an absence, at the same time those cultural features that characterise dialogue have not disappeared and have converged in a new way of encountering and exchange to reshape oneself and welcome the other.

Modern societies, before the COVID-19 pandemic, focused mainly on the body and well-being, largely excluding spirituality and thus narrowing human desires only to the physical sphere. A man of body and emotions dominated over a man of spirit (Kowalczyk et al. 2020). The research conducted as part of the survey indicates that people experiencing fear, suffering or illness often experience a spiritual renewal. Perhaps new people are being shaped, in which the development of spirituality will create a mature attitude based on truth and freedom (Kowalczyk et al. 2020).

*4.3. Women's Experiences of Dialogue during the Blockade*

Structuring an analysis that considers the multiple diversities and meeting points in women's experiences of interreligious dialogue requires an intellectual capacity to understand the complexities of women's subjectivities and their way of creating debate and re-signifying social ties despite the pandemic period. The reflection on the relationship between women and dialogue, which has already been extensively investigated in sociological studies (Ruspini et al. 2018; Canta 2019) is here expanded in relation to the pandemic. Turning our gaze to the social actions triggered by a rupture in people's sociability within religious communities is a need that emerges from below and that also manifested in our research. During our survey of dialogue in the COVID period, we conducted 10 interviews of women opinion leaders willing to share their experiences and tell us about the adaptation strategies of their communities.

Some questions that guided the analysis of the relationship between dialogue, women and the pandemic were as follows. What elements or distinctive features of dialogue can we trace in women's concrete practices and actions? What is the space that women inhabit in the dialogue between religions and in their communities?

Looking at women's subjectivity, their narratives and their positionality (Alcott 1988) for the main elements of dialogue is a total social fact, since this element is intricately intertwined with their private and social life choices. In the interview excerpts that will be presented, it will be possible to identify women's creative capacity to weave social bonds aimed at building spaces for dialogue and freedom.

> Why do you think they are afraid of Women's Freedom? Because [a] woman
> is a bomb. Woman can conquer the whole world; woman would be able to fly!
> (w, RP, Muslim)

Women value certain dimensions of dialogue and consider religious plurality an asset and a result of a process of knowledge of otherness, which produces mutual recognition (Camozzi 2019a, 2019b). The concept of recognition underlies studies on cultural pluralism and Taylor (1989) already used it to motivate the link between culture, identity and society. Unlike men, women have adopted a more informal approach. The former—at least those interviewed—seem more stuck in institutionalized schemes.

There are three distinctive elements in the strategies pursued by the interviewees in their attempt to maintain a dialogue between and within communities: the religious identity, the recognition of otherness, and the propensity to dialogue.

These characteristics emerging from the interviews are intertwined with the theme of the pandemic and its impact on the continuation of dialogue despite the severe limitations on face-to-face relations. The pandemic shocked, surprised, and disrupted all the ancient points of view and certainties, but has given rise to new ones, even in those communities that already made use of the internet.

[ . . . ] we tried to maintain a certain balance. We managed, in fact, as women of
faith in dialogue, between one window and another, to organise a meeting at the
Capitol on mixed marriages. (w, RJ—Coordinator, Jewish)

During the pandemic, the role of women was to propose, at appropriate times, spaces
for confrontation so as not to "fray relationships" because, "I am convinced that a plural
society is a richer society" (w, RP Coordinator, Catholic). The commitment of women
continued to pursue the association's mission, with its plural and polyphonic identity in
the arena of public debate along with other communities.

The public space during the pandemic—except for those possible moments of physical
encounters between people—has remodelled itself on the space-time asynchrony of web
platforms. Religious life has reorganised itself in a domestic space in which the boundary
between private and public has thinned and, in response to the need to give continuity
to each other's spirituality, computers, microphones and webcams were switched on,
dialoguing at a distance.

In fact, the women who took part in the research demonstrated a reactive response to
the restrictions imposed by the global pandemic. The tendency on the part of the women
to draw new experiences, as a response and reaction to the context of restriction, and the
ways in which this process was shaped as needed, was emphasised.

For example, the account of one Catholic woman's experience of the domestic celebra-
tion of religious practices is significant:

The discourse of domestic liturgies was a great offering and a great surprise
[ . . . ]. I can give you the testimony of friends all over the world from Australia
to India, in the meantime, especially we women have experienced the beauty of
being able to be among ourselves as celebrants in our own homes, particularly
for Easter. (w, AWC founder, Catholic)

The ways in which religious discourses and practices have been rearticulated vary
from community to community, just as new forms of dialogue and relationships have
emerged, both within communities and towards the outside world, as these witnesses
point out:

Religion for Peace has found a right way of relating: in addition to the meeting,
there have been prayers in common, in the sense that each one has participated
with their own religious culture in meetings on the theme of, for example, popu-
lations deflated by their rights. So, everyone read their own prayer, their own
psalm . . . and that is also interesting! (w, RP Coordinator, Catholic)

Even though the social distancing of the pandemic has reduced physical proximity, the
situation of discomfort and isolation has also led to a rediscovery of the agency involved in
the celebration of liturgies. For example, the role of women in practicing their faith also
changes from public to domestic space. They have rediscovered a more private dimension
of prayer and a desire to become true protagonists within the community, both as informal
facilitators of dialogue (as has always been the case) and formal leaders with a specific
institutional role. The pandemic has brought to the surface existing problems, such as the
role of women in institutionalized positions within religious communities. Although in
Italy there have been improvements in some communities, such as the Catholic one, with
new appointments, and the role of women being increasingly recognised, there is still much
to do.

My Australian friend said to me: I don't want to think that I'm going back to
church to have to be a spectator! Now, with my husband (a deacon), we celebrate
these liturgies together. That's it! And she rightly said it's bad to think of going
back to simply being a spectator. (w, AWC founder, Catholic)

In analysing the role of women in religious dialogue during the pandemic, the three
concepts already indicated (identity, otherness, and dialogue) guide the narratives of the
interviewees and offer the possibility of understanding the state of health of dialogue
between the different religions and in the communities themselves. In these processes,

women's agency acted as an accelerator at a time of social freezing in both private and public spaces. The experiences of dialogue, therefore, within and between religious communities have not only been reorganised differently, depending on each one's ability to react to the restrictions imposed by the pandemic, but have seen a social and cultural transformation from the customary.

*4.4. Women's Transformation in Dialogue*

While it is true that women have a strong qualitative and quantitative presence, from the testimonies collected we can affirm that the pandemic experience has strengthened the relations within and between different communities, also thanks to the role played by women as facilitators of relations between religions.

Despite recognising the initial difficulties related to the widespread uncertainty caused by the contagions, the interviewees continued to believe that dialogue was still possible. However, resilience and leadership women's skills were evident even during the pandemic, because in a certain way women used to drive the change as active agents. A widespread patriarchal model of religions did not and do not recognise them as having apex roles or decision-making positions. They are consistently considered to be in a gregarious position compared to the men who take the ultimate decision-making power.

In every religion there is a greater presence and prevalence of male roles than female roles; as far as the Jewish religion is concerned, there is a huge difference between the traditionalists and the orthodox and reformed. In the sense that women can occupy the same roles as men! (w, RJ—Coordinator, Jewish)

As Jewish theologists pointed out, there is also a need to shape a new God language, not just the practice; Jewish women have been amending the prayerbook and even structuring whole new services to replace the traditional one because feminist Jews in our time are taking back the power of naming, addressing divinity in our female voices, using language that reflects women's experiences (Falk 1987, p. 41).

With respect to implementing a process of transformation, in general heteropatriarchal power and privilege dynamics are reproduced (Guardi 2017) despite the pandemic. Despite the equalisation in women's rights, their importance also being recognised by some male figures, and the struggles against their discrimination, these traits are often used only to continue to legitimise the male role within religious institutions as the sole representative of these groups. Especially within the religious denominations of the older tradition (although this is not the case for all religions in Italy), there is a different consideration of women that leads to a different treatment, always in favour of men (Zuanazzi 2019).

> Women should be involved for justice and not because we are at the end of our rope [ . . . ] If there are adult women capable of being in communities as leaders, let them do it! [ . . . ] also the fact that things do not come out clearly, that here in Italy we still talk too much about it. So, we will have to do some serious thinking on this. (w, AWC founder, Catholic)

Women have always been the ones who have transmitted the faith in the communities, but this will no longer be the case because they are abandoning it (Zuanazzi 2019). Some of our interviewers also pointed out the problem of female marginalization within religious institutions. Some with a more moderate approach (w, RP Coordinator, Catholic) and some with a more critical one (w, AWC-Founder, Catholic) noted this problem.

Without the rhetoric and avoiding processes of crystallisation and universalisation (Klein 1946), women have a particular resilience and propensity for organic solidarity. In the experiences reported, shared initiatives become central to the interviewees' commitment—from prayer times (together, but distinct) to services for the most fragile, from cultural activities to attending to mixed families. The women of different religions who meet tend to encourage each other and strengthen their emancipatory trust within their own communities. Moreover, women themselves often act as bridges and become de facto reference points for intercultural and interreligious dialogue in their daily lives. Their

presence is limited in formal meetings, but they always work for encounter and dialogue (w, Cipax-President, Catholic).

> Because women are more open, I think, less dogmatic, they have the audacity, they have the soul, the spirit that flies! And I believe they have the concept of welcoming, they welcome truths, they have this capacity to welcome. Diversity, that is. (w, Religion for peace, Muslim)

This role of weavers of relations in dialogue is often underestimated or disregarded when hypotheses of change are presented in the public space, precisely because it is difficult to renounce privilege and power, instead redefining the position of women in dialogue and in the care of encounters. Women remain confined to a purely executive space within religious institutions, never creative or authoritative, except within women's religious communities where, still, men hold the pinnacle. Caring practice "is often overlooked and can remain invisible or seem meaningless in society, as it is based on qualitative rather than quantitative values" (Vaughan 2005, p. 33).

## 5. Conclusions

Dialogue survived in personal and rooted relationships during the lockdown: it remained under wraps and was then resurrected in the next stage. This is the first result that appeared from the analysis of the interviews. There is a relationship between how the communities lived their faith, and how they promoted interreligious dialogue. It is also noted at the level of religious institutions when together they promoted the Day of Prayer for Humanity celebrated during the lockdown. This behaviour is also found among our key informants. The temporary emergency of the pandemic crisis shows a characteristic towards which the representatives of the various religions are heading: to offer interpretations to questions of meaning about life and death. The choice of dialogue is to recognize each others' legitimate ways, through which each person can set out on the way to the Truth.

Although the physical meetings could not take place, the interviewees said there were no lack of moments of prayer and communion. In response to the need for dialogue, all the communities employed social networks that represented a bridge during the pandemic period to feel close in the distance and present in the absence (Parish 2020). Although many moments could not have been replicated—such as shared folk festivals (PIRS—member) or intercultural trips that transformed dialogue into a concrete act (w, TVI—President; ERPN—Coordinator)—moments of shared prayer were not lacking. They allowed the religious communities to continue the dialogue and, especially, to pray together for the end of such a dark period. Other groups have reorganized new moments of dialogue that they could not have imagined. The Friday Prayer, for example, moved online, opening up to men and women and breaking down many barriers; the Eid al-Fitr—the end of Ramadan—transformed from a moment of dialogue and celebration only for the Muslim community into a true act of dialogue between different communities, but online. The pandemic created a kind of shared experience that communities were able to reflect on together and has forced us to reconceptualize some religious practices consistent with the plural nature of our time (Seligman and Weller 2012), even if it did not stop the desire to reorganise the dialogue through a physical presence.

Some initiatives were proposed by the institutions, while others spontaneously arose within the groups of believers. In the latter case, the women were the protagonists and took an active role in proposing rituals and encounters. In such situations, the image of women following religious men (as priests or imams) has been overturned. The dialogue initiatives undertaken do not arise on neutral ground in Italy. A characteristic emerges from the interviews, within an overall climate of openness to dialogue: the positional advantage of the Catholics, compared to the other communities present. The advantage translated into a greater response capacity of the religious institution compared to others (such as through the presence of ecclesiastical representatives who could provide spiritual assistance in hospitals or prisons). However, this advantage in an atmosphere of openness to dialogue

creates a disparity which can lead to fissures in relations. This is another element that the COVID emergency has clearly demonstrated. The recognized disparity of treatment between the different religions can be an occasion to strengthen the request to formulate an effective law on religious freedom.

Finally, a female-specific characteristic can be observed in the women's interviews. They appear to be weavers of bonds (Toki et al. 2015), even from a distance. The themes that recur in the analysis between the role of women, religious dialogue and pandemics are characterized by their placement at the centre of the concrete actions of cultural pluralism, based on certain concepts used in a dynamic manner: religious identity, the recognition of otherness, and the propensity to weave virtuous links with encounters and words. Although a position of marginality still emerges, especially when looking at the top positions, in practice, women's actions move small changes to transform a typically-heteropatriarchal way of conceiving roles and hierarchies (Bettie 2006; Rahemtulla 2017).

In dialogue, they are aware of their identities and belongings and for this reason, they are promoters of encounters with the other. The relationships that arise from interreligious dialogue cross the boundaries of individual communities and on the one hand feed the innovative creativity of women (Knibbe and Bartelink 2019), while on the other, becoming a stimulus to emancipate from the patriarchal models of religious institutions (Zwissler 2012). Above all, the social bonds that women in interreligious dialogue build through their memberships, actions and forms of religiosity can be configured as a peaceful revolution (Vaughan 2018): a revolution that has within it a factor that drives social change and transformation. Women who live on the margins both in the "feminist" field and in their faith communities (Nyhagen 2019; Giorgi 2021) carve out a particular role for themselves with their engagement in interreligious dialogue. It becomes an outpost in which to experiment with a new female figure who is not yet recognized in their communities and contribute to the construction of models of peaceful coexistence.

**Author Contributions:** Conceptualization, A.C.; methodology, A.C.; formal analysis, C.C. and A.F.C.; investigation, A.C., C.C. and A.F.C.; resources and data curation, A.C., C.C. and A.F.C.; writing—original draft preparation, all the authors shared the research process and the results obtained, in particular A.C. wrote Sections 1–3. A.F.C. wrote Sections 4, 4.1 and 4.2; C.C. wrote Sections 4.3 and 4.4. all the authors wrote the Conclusions; writing—review and editing, A.C., A.F.C. and C.C.; supervision, A.C.; project administration, A.C. All authors have read and agreed to the published version of the manuscript.

**Funding:** This research received no external funding.

**Institutional Review Board Statement:** The study did not require ethical approval.

**Informed Consent Statement:** Informed consent was obtained from all subjects involved in the study.

**Data Availability Statement:** Data is unavailable due to privacy.

**Conflicts of Interest:** The authors declare no conflict of interest.

## Note

1    For further information (Caritas 2021; Cesnur 2021; for a critical analysis see Pace 2013b).

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
