# Peer review of "Living Interfaith Dialogue during the Lockdown: The Role of Women in the Italian Case"

_religions, doi:10.3390/rel14020252_

Round 1
Reviewer 1 Report
The study presents how during a global pandemic (covid-19), which led to a lockdown in Italy, the inter-religious dialogue was made possible thanks to the mediated media/technology of online video meetings. These meetings led to some changes in the nature of the dialogue and a challenge to the gendered division of labor practiced in the religious nature of the meetings.
The study deals with an important and interesting topic that expresses how in unstable times and with the mediation of the media, new forms of inter-religious meetings are organized. However, additional work is required to be ready for publication.
In general, the study in its current form lacks a more comprehensive literature review on women, religion, and media, on the role of technology or media in encounters between different social and religious groups, and more on types of dialogue between groups and people, especially in the theological field.
Also, there is an additional lack of reference to the contexts - to the need for inter-religious dialogue, to what happened in Italy before the COVID-19 epidemic in the field of inter-religious dialogue, and in general to the need for such meetings.
A method of analyzing the interviews and a description of how the analysis was done are missing.
Also, it is useful to reduce the length of the quotations. It is important to explain the quotations presented, to connect them properly, and to explain what each quotation adds to the other.
Also, there is a lack of significant and relevant theories about communication, media, and religion that can help explain the main findings.
I find that significant additional work is required but is certainly possible. I encourage the authors to do so for the sake of a quality and important article.
Below is my more focused reference
Abstract
· It is worth mentioning explicitly that this is Covid 19 and mentioning already in the second line that this is a case study that took place in Italy
· The findings are not presented in the abstract
Introduction
· An explanation of the relations between religions in Italy and information about their situation (including statistics) is required
· It is also worth introducing the period - what happened in Italy during the plague and how it relates to religious matters
Literature review
· There is a lack of reference to the research literature on unstable situations and insecurity such as terrorism and wars and their relationship to religion and inter-religious relations from a historical and mainly sociological perspective
· Likewise, the place of women in religion and their place in meetings and dialogues, in renewing and challenging traditions and conservative concepts, must be presented. In this field, there is extensive literature on religious feminism. To this, it is important to add the use of media, as you present in the article. An example of a different case but similar in characteristics were presented in the article: "The stranger within: Israeli religious women lead social change in community web-series".
· In defining types of dialogue, it is important to also review the ethical, social, and theological aspects, as expressed in the approach of both Martin Buber and Emmanuel Levinas to dialogue. In these cases, there is no place for gender or religion, rather it is an inter-subjective and Interfaith encounter that leads to a shared theological-divine revelation. See for example the book: " Martin Buber: The Life of Dialogue" and the article: "Levinas and Interfaith Dialogue". I think the book "Dialogic Moments: From Soul Talks to Talk Radio in Israeli Culture" will also contribute to the review on dialogue.
Methodology
· At the beginning of the methodology, it is useful to return and present clearly the goals of the research and the research questions. It seems that it is worth adding a question that was not presented in the introduction and it is expressed in the first theme, which does not deal with dialogue at all, as I will also mention later. Another research question on the role and place of women in media-mediated dialogue should also be added.
· It should be explained in relation to the key informants chosen for the interviews - are these the main or the only people who actually dealt with inter-religious relations in Italy because Buddhists and different factions of Christians such as Protestants are missing. It is also worth noting among Muslims whether they are Shiites or Sunnis. Also, it is important to present the ages of the interviewees.
· The methodology is missing a very important part - the analysis method. How were the interview transcripts analyzed?
· There is no explanation of the advantage of this data collection method (the interviews) over other methods that were not used in the study such as observations and content analysis.
Results
· The first theme does not deal with dialogue, therefore, as I mentioned above, a research question on the use of digital/media should be added.
· Following the quotations presented, it is important to explain what is implied by the words of the interviewees. That is, the claim must be analyzed.
· In presenting the use of digital, emphasize what the interviewees thought it led to - to what change in the religious context? Has it become more individual practices? For more modernity or open to modernity? That is, is the digital only a tool for meeting/dialogue or does it change the perception and thinking about religious practices?
· Among Muslims, you seem to have found a change in gender accessibility to religious prayer - is it and how is it expressed in other religions?
· Throughout the article you present meetings between people as a dialogue. This is not something coincidental. There is a meeting and there is a dialogue with different characteristics and results. You must present with the help of empirical materials how the dialogue is expressed. For example, in the first theme in paragraphs 290-293, you state that several religious groups have participated in the online space with the Muslims and this is a dialogue - but this is not empirically validated in quotes or other references - that it is a dialogue according to its various characteristics.
· In the second theme, the interviewees' quotes about religious freedom and inequality do not contribute to the issue of religious dialogue, which is the main issue in this theme. Therefore, the whole discussion about inequality should be presented briefly as part of the field and from here go out and present how dialogues relate to this as well and how it is related to unstable times/epidemic.
· In connection with an inter-religious and musical meeting, it should be clarified how this constitutes a vocal/sensory or verbal dialogue - and what digital mediation does - is there no creativity and thinking outside the box if the physical meeting is absent? As shown in paragraph 418 and before, it is claimed that there is still a dialogue, but no reference is made to the type, form, and content of the dialogue and its consequences. That is, there is no reference to the dialogue itself, but the engagement remains at the level of an encounter and not a dialogue.
· At the end of this theme, in paragraphs 441-435, there is a paragraph that is not empirically proven.
· In the third theme, which deals with the role of women, it is not clear how the female-gender identity and accordingly the social role of women in the religious organization is what enables a certain dialogue, and it is not clear what its characteristics are. How is it different from men's practice?
· It is noted that there are new forms of dialogues and relationships, but these are not presented and explained theoretically in relation to gender, religion, and the place of women in religion.
Conclusions
· There is a lack of theoretical reference to the findings - to the role of technology/media in meetings and unstable times for gender and religious changes.
Author Response
Dear reviewer,
thank you for your suggestions and comments
we highlight the changes we have inserted to enrich the article according to your proposals.
We have tried to add what was reported missing in the article
We have rearticulated the introduction to better explain the topics of the article, We have highlighted what happened in Italy during covid. Instead, we highlighted - thought - to give indications on the Italian religious context in the methodological paragraph.
In paragraph “How can we define dialogue” we have deepened the theme of interreligious dialogue and the female/feminist dimension. We have enriched the definition of the dialogue also referring to the works of Levinas and Buber as suggested. We didn't talk about terrorism, but about contributing to peace through the citation of two articles, one talks about interreligious dialogue between a community of Reformed Jews and Muslims and one talks about the contribution of women of different religions to peacemaking in Nigeria.
We explained in the methodological part what were the advantages and disadvantages of interviewing opinion leaders online. We also pointed out that it was the only way to interview them at the time.
In the analysis of the results, we reorganized the topics covered. We have tried to answer the various questions both on the impact of "media coverage in the community" and on some changes that have taken place - in some cases relating exclusively to the time of the lockdown. Interview quotes have been shortened and comments expanded, as suggested. We have tried to clarify the limits to dialogue that derive from the lack of a specific law on religious freedom in Italy. We have inserted empirical evidence in the written statements and we have summarized some parts.
In the conclusions we have included some theoretical references
We hope we have clarified what you requested
Reviewer 2 Report
This article is based on interviews conducted about inter-religious dialogue during the pandemic in Italy. The paper aims to give an account of the challenges faced and strategies employed by religious actors as they were forced to transition to web-based modalities during the COVID-19 pandemic. The main claims seem to be that 1) despite the social distancing, religious leaders continued to put stress on and succeed in engaging in meaningful inter-religious dialogue and 2) women have played a particularly important role in maintaining social bonds and inter-religious bonds during the pandemic.
The topic of the article is important and significant. While most literature thus far has focused on how people within specific religious traditions have responded to the pandemic, this article aims to understand the impact on inter-religious ties and dialogue. [As a side note, the title could be more descriptive and reference the pandemic and Italy; keywords could also be updated to get more visibility in search.] In many places, the pandemic has exacerbated existing inter-religious tensions, and drawing out these instances could be helpful to underscore why this case study is important. In addition, it would be helpful to reflect more on the specific experiences of the pandemic in Italy and what the insights (and also limitations) are from the situatedness of this particular study. It would also be important to reflect on the insights and limitations of the methodologies chosen. First of all, the choice to focus on interviewing religious leaders could be important in getting information about interfaith initiatives and dialogues. Yet, it is also a decision to focus on elites who were likely already active in dialogue before the pandemic. It would be useful to reflect on this in the text of the article. In addition, it is very possible that women played an important role in inter-religious dialogue, yet it is also important to ask whether this conclusion comes from the fact that 2/3 of interviewees were women. It remains unclear to me whether the researchers also attended any inter-religious online events or whether the analysis is taken exclusively from the interviews.
The argument indicates that despite the challenges of the pandemic, religious dialogue continued and was even able to thrive as meetings shifted online. In this analysis, it might be important to take the impact of the modality into consideration to understand how being divided into boxes on a screen during a zoom meeting might impact the way people can connect across religious beliefs—creating certain kinds of ‘commensurability’ and ‘incommensurability’ (Mair and Evans 2015 “Ethics across borders: incommensurability and affinity”). Or was it effective because the pandemic created a kind of common experience that communities were able to reflect on together (for more on this, see Seligman and Weller 2012- Rethinking Pluralism)? It seems the argument indicates that many of these meetings were continuations or in-person relationships that had already been established, rather than new bonds which were forged. Does this mean that having that previous in-person experience was still central in inter-religious dialogue during the pandemic?
In terms of organization, the text could benefit from some editing. At times, data/quotations involving interviewee’s reflections on practicing their own religious traditions during the pandemic are given. While these are important or interesting, they are often mixed in with quotations about inter-religious dialogue. In addition, the article includes many long quotations from interviewees. To some extent, this can be helpful to give a sense of the conversations and the texture of responses—but it must be balanced with the analysis rather than standing in for it.
I would like to raise concern about some classifications and unqualified statements throughout the article. I would like to question the decision (p. 5) to group religions as “monotheistic religions”, “oriental religions” and “new religious movements” and whether this might be unnecessarily reinforcing a West/East dichotomy that might not be reflective of the transnational flows of religion and the way they are lived in Italy today. It is also unclear why Bahai is classified as a “new religious movement”. There are also several unqualified statements throughout the text that appear as problematic without any explanation or citation. Some examples:
p. 12: “Modern societies, prior to the Covid-19 pandemic, focused mainly on the body and well-being, largely excluding spirituality and thus narrowing human desires only to the physical sphere”
p. 13: “Women value certain dimensions of dialogue and consider religious plurality as an asset and as a result of a process of knowledge of otherness, which produces mutual recognition”
Author Response
Dear reviewer,
thank you for your suggestions and comments, and thank you for appreciating our proposal to investigate interreligious dialogue in times of the pandemic.
we have changed the title and the keyword according to your proposals. Then we specified that this is an Italian case study and we underlined the theme of the Covid 19 in the introduction. We have also tried to give more value to the part on the contribution of women to interreligious dialogue.
We explained in the methodological part what were the advantages and disadvantages of interviewing opinion leaders online. We also pointed out that it was the only way to interview them at the time.
In the analysis of the results, we reorganized the topics for higher clarity. We have also highlighted that the meetings in question were above all continuations of in persons meetings. In our opinion, the first result worth mentioning is precisely the ability to maintain a "fragile" bond even during such a particular time of crisis. Interview quotes have been shortened and comments expanded, as suggested.
We decided to eliminate the classification of religious groups because it rightly needed more insights, but they were not central to the objectives of the article and would have required too large a space for description. However, we have included some bibliographic references on the Italian specificity. In the other cases an attempt has been made to clarify the statements. We hope we have clarified what you requested
Reviewer 3 Report
REVIEW
I suggest to accept this manuscript under major revisions:
1. Introduction: the argument is not clear. There are too many questions, instead of clear argument. Rewrite this section and try to convince your readers how by this micro issue you can tell us something more broadly about contemporary religion\intersection of gender-religiosity.
2. Theoretical background: There is a lack of literature about sociology and anthropology of interfaith encounter. The section "How can we define dialogue" - should be centralized the issue of interfaith dialogue – and emphasize the gender impact. Please see for instance: Ben-Lulu, E., & Feldman, J. (2022). Reforming the Israeli–Arab conflict? Interreligious hospitality in Jaffa and its discontents. Social compass, 69(1), 3-21.
Furthermore, "We can therefore extrapolate three key elements that form the framework for the analysis of the interreligious dialogue of research" – Why? Why not choosing in Grounded Theory and classify the findings after analyzing them? Why to fit the theory to the field and not choosing the opposite way?
3. Methods: please explain more of advantages and disadvantage of virtual\remoted research. In addition, what's your positionality in the field, please add a reflexive comment.
4. Analysis: I suggest to reorganize the classification of the findings to three\four themes, because some of the issues are relevant to different sections. Try to think about some gender\religious categories which specify and emphasize the women's' role. The current of classification – by type of worship – doesn’t really encourage and support your argument. Please engage with other researchers who demonstrates how categories of sacredness\ holiness\ embodiment\ materialism are part of creating of remoted interfaith encounters. For example:
Parish, Helen. 2020. The absence of presence and the presence of absence: Social distancing, sacraments, and the virtual religious community during the COVID-19 pandemic. Religions 11(6): 276
Scott, S. AQ. 2016. Algorithmic absolution: The case of catholic confessional apps. Online-Heidelberg Journal of Religions on the Internet 11: 254–275.
Ben‐Lulu, E. (2021). Zooming In and Out of Virtual Jewish Prayer Services During the COVID‐19 Pandemic. Journal for the Scientific Study of Religion, 60(4), 852-870.
Heilman, Samuel C. 1976. Synagogue life: A study in symbolic interaction. Piscataway, NJ: Transaction Publishers.
Helland, Christopher. 2000. Religion online/online religion and virtual communitas. In Religion on the Internet: Research prospects and promises, edited by Helland, Christopher, Jeffrey. K. Hadden, and Douglas Cowan, pp. 205–24. Amsterdam: Elsevier Science Inc.
5. Discussion: Provide more conclusions about the intersections of feminine religious leadership in anomaly time. Try to think about it by adopting feminist theology\liturgy landscape. Maybe Falk or can contribute to your argument and help you to explain the women role\position in this matter. Here some references:
Falk, Marcia. “Notes on Composing New Blessings toward a Feminist-Jewish Reconstruction of Prayer.” Journal of Feminist Studies in Religion 3:1 (1987), 39–53.
Walton, Janet Roland. Feminist Liturgy: A Matter of Justice. Collegeville, MN: Liturgical Press, 2000.
Author Response
Dear reviewer,
thank you for your suggestions and comments
we highlight the changes we have inserted to enrich the article according to your proposals.
We have rearticulated the introduction to better explain the topics of the article
In paragraph “How can we define dialogue” we have deepened the theme of interreligious dialogue and the female/feminist dimension.
Moreover, the three key elements that concluded the paragraph had been indicated to accompany the reader in reading the results. They have been removed, since the impression was given "to fit the theory to the field". This was not our aim.
We explained in the methodological part what were the advantages and disadvantages of interviewing opinion leaders online. We also pointed out that it was the only way to interview them at the time.
In the analysis of the results, we reorganized the topics covered and took advantage of the good articles suggested. However, our analysis fails to categorize women's different roles, most likely because we focus on opinion leaders who hold similar attitudes. Instead, we have identified - and highlighted especially in the conclusions - some gender aspects that seem interesting to us for the contribution to dialogue.
We hope we have clarified what you requested
Round 2
Reviewer 1 Report
The manuscript in its current version has been greatly improved and only a few additions remain to allow its publication.
Below are my comments on the required additions:
First, the methodology still lacks a research method. Interviews are a data collection method, but it is not an analysis method. A method, such as thematic analysis, is required to narrow down the issues that have arisen. But this requires an explanation of the process that makes it possible to extract the themes.
Second, a more significant reference to feminism and religion is still missing - both in the literature review and in the concluding discussion. In the literature review, there is a reference to a relevant source, but this is not enough. In the concluding discussion, a more significant concern about the findings and their connection to feminist theory in the religious context is required. That is, a theoretical discussion is required that links religious feminism to your findings.
I wish the writers the best of luck.
Author Response
Thanks for your comments and suggests
We have described our methodology now: ). we write: "To interpret the answers to the interviews, an epistemological approach was adopted to understand the social construction of the observed phenomenon (Della Porta 2014). Through a content analysis framework the statements collected by the interviewees were analyzed (Cipriani et al. 2021)."
We have expanded the references on the theme women-feminism-religions both in the description of the theoretical framework and in the conclusions
Best regards
Reviewer 2 Report
The topic of the article is quite important and will be an important contribution to our knowledge about the impact of the pandemic on inter-religious dialogue, an underexamined aspect of the pandemic thus far. That being said, there are a few aspects of the article that could still be improved prior to publication.
1. Introduction: The article mentions the promotion of inter-religious dialogue during crisis as if it is a natural outcome. Yet, as we know from historical pandemics like the Black Death (and subsequent persecution of Jewish populations) , they can also be used to scapegoat minority religious and/or ethnic groups. During COVID-19, there were also numerous cases of inter-religious tensions which flared up, including Indian Muslims (Tablighi Jamaat) being accused of performing “corona-jihad” to infect non-Muslims. In the Italian case, it seems that there were many cases of anti-Semitic conspiracy theories about the virus itself. I point this out because it is important to recognize that in these cases of inter-religious dialogue take place in a specific and situated context which may further underscore the need for dialogue to occur (but that it is not necessarily a natural outcome). It is important to know how/whether this shaped the dialogues taking place or if it came up at all in respondents’ evaluations of inter-religious dialogue at the time.
2. There are discussions about the way in which religious practitioners experienced the impacts of the pandemic on their own religious practice and within their own religious tradition. While they are extremely interesting, a clear link needs to be made in terms of how this impacts the main subject of the article: inter-religious dialogue.
3. Role of women: The addition of sources has helped to clarify further and justify the focus on women and their specific role in inter-religious dialogue. However, there seem to be a priori assumptions about women as necessarily moderate and promoting peaceful dialogue. While many of the citations provided do show case studies where women have taken on important roles in inter-religious dialogue, the data needs to also be clearly explained and analyzed to demonstrate how and why this case study also indicates women are playing a similar role.
4. The article needs another round of in-depth editing for English style/language
5. Some of the in-text citations are missing from the bibliography
Author Response
Thank youfor appreciating our research topic.
- In the introduction we have included some observations to highlight the Italian context. We underlined the media risk of anti-Semitism, but above all the debate that emerged on the limitation of religious freedom and the protests of the no vax movements.
- We try to explain in the analisys of results the link between internal dialogue and interreligious dialogue: "Internal communitarian dialogue is also the first step that subsequently leads to interreligious dialogue: it enables the sharing of experiences and values within a community to open up to the understanding of others".
- We have extended the observations on the role of women in interreligious dialogue, especially in par. 4.3 and 4.4
Reviewer 3 Report
attached my review

Author Response
Thank you for enjoying the topic of our article.
Introduction: We have tried to include the attention required by trying to match the other suggestions offered to us as well.
Theoretical background: We have expanded the references on the feminism relationship Thanks above all for the reading advice of Rahemtulla, S. (2017). Qur'an of the oppressed: liberation theology and gender justice in Islam. Oxford University Press-Religion. We did not know the text - very interesting.
Method. we have briefly added our reflective position on the choice of the qualitative method.
Analyses.
a. We have tried to clarify our findings better.
b. We have already introduced different parts in the article about some characteristic in Italy. For example:
Paragraph 4.1. Seek out new ways for ancient practices
In some religious communities in Italy, such as the Catholic ones, and different age groups during the pandemic, religious feelings increased compared to the pre-Covid period (Zigulsky 2020). The importance of the Word of God and prayer, and the need to give meaning to one’s life at a time when everything that previously made sense has vanished, have increased; religious practices at home such as prayer, reading the Gospel or devotions have also increased.
The importance of the Word of God and prayer, and the need to give meaning to one’s life at a time when everything that previously made sense has vanished, have increased; religious practices at home such as prayer, reading the Gospel or devotions have also increased (ivi).
Paragraph 4.2. Interfaith dialogue in Italy
First of all, of particular relevance is the issue of religious freedom in Italy that necessarily emerged during this period of emergency. Religious freedom is recognised as a constitutional right but lacks a framework of law and a passage on the different starting points of religions. Each community had different spaces of freedom due to restrictions that created inequalities and difficulties to proceed with the interreligious dialogue between each other, particularly for those communities who did not sign an agreement with the State (f, TVI – President; ERPN - Coordinator, Tathata Vridham).
This situation has produced a difference in positions between communities in Italy, which remained latent before the pandemic, and a sense of revenge against the majority religion, harshly reported in particular by the atheist interviewees (f. UAA – former president) and also recognized by Catholics as an impediment to a peaceful dialogue (m, IEC, Catholic). Despite the presence of Christian forms other than Catholicism or other religious forms, Italian culture continues to be imbued with Catholic values in several areas (Cesnur 2021). After the first lockdown, when the different places of worship began to open again, a different procedure emerged between communities that had an agreement with the State and those that had not.
Paragraph 4.3 Women experiences of dialogue in lockdown
Although in Italy there have been improvements in some communities, such as the Catholic one, new appointments, and the role of women has been more recognised, there is still much to do.
Paragraph 4.4. Women’s transformation in dialogue
Especially within the religious denominations of the older tradition, although this is not the case for all religions in Italy, there is a different consideration of women that leads to a different treatment always in favour of men (Zuanazzi 2019).
Conclusion: We have expanded our conclusions in the requested direction, and indeed we feel they have improved thanks to the suggestions.
Best regards